# Drug Safety During Breastfeeding: A Comparative Analysis of FDA Adverse Event Reports and LactMed^®^

**DOI:** 10.3390/ph17121654

**Published:** 2024-12-09

**Authors:** Hülya Tezel Yalçın, Nadir Yalçın, Michael Ceulemans, Karel Allegaert

**Affiliations:** 1Department of Toxicology, Faculty of Pharmacy, Hacettepe University, 06100 Ankara, Türkiye; hulya.tezel@hacettepe.edu.tr; 2Rega Institute for Medical Research, KU Leuven, 3000 Leuven, Belgium; 3Department of Clinical Pharmacy, Faculty of Pharmacy, Hacettepe University, 06100 Ankara, Türkiye; nadir.yalcin@hacettepe.edu.tr; 4Clinical Pharmacology and Pharmacotherapy, Department of Pharmaceutical and Pharmacological Sciences, KU Leuven, 3000 Leuven, Belgium; michael.ceulemans@kuleuven.be; 5Child and Youth Institute, KU Leuven, 3000 Leuven, Belgium; 6Research Foundation Flanders (FWO), 1000 Brussels, Belgium; 7IQ Health, Radboud University Medical Center, 6525 XZ Nijmegen, The Netherlands; 8Department of Development and Regeneration, KU Leuven, 3000 Leuven, Belgium; 9Department of Hospital Pharmacy, Erasmus MC, 3015 GD Rotterdam, The Netherlands

**Keywords:** breastfeeding, adverse event, safety, infant, FAERS, LactMed^®^

## Abstract

Background/Objectives: While breastfeeding is highly recommended, breastfed infants may be exposed to drugs by milk due to maternal pharmacotherapy, resulting in a risk of adverse drug events (ADE) or reactions (ADRs). The U.S. Food and Drug Administration (FDA) Adverse Event Reporting System (FAERS) is an online pharmacovigilance database, while the Drugs and Lactation Database (LactMed^®^) includes accurate and evidence-based information on levels of substances in breast milk and infant blood, and possible ADRs in nursing infants. We aimed to explore the FAERS database and compare ADE/ADR information patterns between both databases. Methods: The FAERS database was explored (29 July 2024) for ADEs related to drug exposure during lactation to determine annual trends, infant outcomes, and regions of reporting. The active pharmaceutical ingredients (APIs) associated with these ADEs were categorized based on the Anatomical Therapeutic Chemical (ATC, first level) classification. The top five APIs in each ATC group were explored in terms of the type of ADEs reported and compared to information in LactMed^®^. Results: In total, 2628 ADEs were obtained from the FAERS database, with increased reporting over time. In the FAERS database, 68.4% of the patients were under 2 months old, 5.5% had life threatening ADEs, and 3.6% died, while 84.70% of the cases were categorized as serious. Most ADEs were from North America (44.9%). Most drugs (50.9%) were nervous system drugs. The most frequent reported outcome was “other outcomes (without additional subdivision or information)” (58.2%), reflecting the diversity in outcomes reported. When related to the same drug, the FAERS database and LactMed^®^ resource exhibited both similarities and differences in the types of reported ADE/ADR. Conclusions: The FAERS database is a useful tool to detect potential ADEs (rather sensitive), without ADR assessment, while LactMed^®^ provides guidance driven by relevant ADRs (rather specific). The FAERS database is useful to obtain exploratory information about ADEs during lactation to increase the knowledge about drug safety during breastfeeding and the awareness of the possible risks in nursing infants, while LactMed^®^ translates all available information into guidance.

## 1. Introduction

Exclusive breastfeeding is advised for the first six months after delivery by the World Health Organization (WHO) and the American Academy of Pediatrics (AAP) [1,2]. It is very well known that breastfeeding provides advantages for both the mother and the breastfed child (Figure 1) [2,3].

Although there are differences in geography and demographics, about 90% of women, at present, initiate breastfeeding because of improved awareness about the advantages of breastfeeding [2,3]. However, investigations based on population studies showed that over 50% of nursing mothers use prescribed drugs [4]. While a mother’s milk obviously provides health benefits to the newborns or infants (Figure 1), it may also expose them to potential risks from drugs that are not meant to treat conditions in the infant, for example, when a breastfeeding woman takes a potentially harmful drug, which may appear in clinically significant amounts in breast milk [5].

In general, drugs can transport to breast milk by passive diffusion from maternal plasma and across the mammary epithelial cell, by carrier-mediated transport from the maternal plasma, lipid co-transport, and transcytosis [1]. Drug levels in milk could be explained or estimated by considering several pharmacokinetic and physicochemical characteristics, such as—but not limited to—drug clearance, milk-to-(maternal) plasma concentration (M/P) ratio, or relative infant dose (RID). RIDs are helpful in risk assessment as they show the amount of the drug that a baby consumes through breast milk divided by the mother’s drug dose (corrected for maternal and infant body weight). Further, the M/P ratio based on the area under the curve (AUC) concentration needs to be evaluated in conjunction with maternal drug clearance value and bioavailability [4].

In clinical practice, the absence of sufficient data on the amount of drug passage to breast milk makes it challenging to determine the advantages and risks of pharmacotherapy in breastfeeding or to support shared decision processes [5]. While the majority of drugs taken by and studied in lactating mothers do not appear to have a clear negative impact on the nursing child, case reports have demonstrated instances of severe and serious infant events during breastfeeding [4,6,7,8].

An efficient pharmacovigilance (PV) system would be a relevant asset to proactively monitor the safe use of drugs to promote public health [9]. The U.S. Food and Drug Administration (FDA) Adverse Event Reporting System (FAERS) [10], the European Medicines Agency (EMA) EudraVigilance [11], and the Australian Therapeutic Goods Administration Database of Adverse Event Notifications (DAEN) [12] are online pharmacovigilance databases including information from reports of adverse events. In contrast, the National Library of Medicine (NLM)’s TOXNET system (https://www.nlm.nih.gov/toxnet/index.html, accessed on 29 September 2024) generally presents the toxicity and safety profiles of chemicals, and evidence on environmental health. The data source LactMed^®^ is part of the TOXNET system. This online source offers up-to-date, evidence-based information about the concentrations of drugs and other chemicals in breast milk and infant blood; possible adverse effects in the breastfed infant; and alternative pharmacotherapeutic options [13].

Obviously, case reports and series about the occurrence of adverse drug events (ADEs, time-dependent) or adverse drug reactions (ADRs, causality assessment included) in nursing infants have been published. However, this literature is rather scarce, and no clear evidence is available on the level of agreement between spontaneous reports of ADEs/ADRs available in pharmacovigilance databases and the knowledge about infant adverse effects during breastfeeding in reliable and point-of-care resources aimed at healthcare professionals. Therefore, this study aimed to describe the pattern of ADEs reported in the FAERS database, and to compare the number and type of ADE reports in infants related to breastfeeding to the information on side effects in the lactating infant as reported in the LactMed^®^ database.

## 2. Results

### 2.1. Number of Lactation-Related Adverse Events and Annual Trends

In total, 2675 lactation-related case reports were identified in the FAERS database for infants under the age of 2 years. Since the patients’ age and weight information were not stated in 47 cases (1.7%), these reports were excluded from the analysis. As a result, the final number of reports included in the study was 2628. In the FAERS database, 68.4% of the patients were under 2 months old. The full dataset has been provided in Appendix A. The reporting trends over the years, between 1 January 2001 and 31 March 2024, are shown in Figure 2. Over the study period, there has been a continuous increase in reports between 2001 and 2019, followed by a somewhat lower annual number between 2020 and 2023 (for 2024, only data for the first 3 months were available).

### 2.2. ATC Categories and Most Commonly Retrieved Active Pharmaceutical Ingredients

Most of the reported drugs in the lactation-related reports (50.90%) belonged to the nervous system (N). Moreover, 19.90% were antineoplastic and immunomodulating agents (L), 7.20% were anti-infectives for systemic use (J), 7.20% were related to alimentary tract and metabolism (A), and 3.40% were respiratory system treatments (R). Figure 3 provides an overview of the distribution of the APIs involved in lactation-related reports according to the first ATC level.

The five most commonly reported APIs within each of the five most prevalent ATC classes are presented in Table 1.

### 2.3. Comparison of the ADEs from the FAERS Database to the LactMed^®^ Database

Appendix A provides an overview of the information on adverse infant events in the FAERS and LactMed^®^ database. For APIs related to the nervous system, the description of the AEs seemed to relate to opioid-receptor activation mechanisms (buprenorphine), but with other clinical descriptions or terminology. In contrast, for the events related to acetaminophen, this reflected differences between time-related events and causal reactions. For the antineoplastic and immunomodulating agents, the absence of reports for adalimumab, infliximab, and tacrolimus in LactMed^®^ was observed, while a diverse list of time-related adverse events was retrieved from the FAERS dataset. A similar pattern was observed for the ATC level alimentary tract and metabolism with no reports in LactMed^®^ for insulin, omeprazole, ondansetron, and metformin, or the ATC level respiratory system with no reports in LactMed^®^ for omalizumab, fluticasone, and budenoside. For the APIs related to the ATC level anti-infectives for systemic use, a rather diverse and heterogenous pattern was noted, for example, with respect to the gastro-intestinal side effects reported in both databases.

### 2.4. Outcome Categories of Lactation-Related ADEs in the FAERS Database

The distribution of the infant outcome categories among the lactation-related ADEs in the FAERS database is provided in Figure 4. While 2264 (84.6%) of them were stated as serious cases, 106 (4.6%) were reported as death cases. The most frequent outcome was stated as “other outcomes, no additional detailed information provided” (58.2%). This was followed by “hospitalized” (38.1%) and “non-serious” (15.3%) outcomes.

### 2.5. Regional Origin of the Lactation-Related Adverse Events Retrieved in the FAERS Database

The distribution of lactation-related adverse event reports by continent is shown in Figure 5 and Figure 6. Most reports originated from North America (44.9%), followed by Europe (27.2%) and Asia (8.1%).

## 3. Discussion

### 3.1. Main Findings

This study aimed to describe infant adverse events reported as part of lactation-related ADEs in the U.S. FAERS database, and subsequently compare the available information on type and patterns of ADEs in the FAERS database to ADRs reported in LactMed^®^. Overall, 2628 FAERS reports were included in the study.

Related to the first aim (FAERS pattern), we observed a fluctuating but increasing trend in the number of reports over the years, despite somewhat of a decrease in the most recent years (Figure 2). Most lactation-related reports in the FAERS database originated from North America (44.9%), followed by Europe (27.2%), and Asia (8.1%). While ”other outcomes” was the most commonly reported outcome as a category, the dataset did not provide additional information to further explore this large group of events (Figure 2).

According to population-based data, more than 50% of nursing women took prescribed drugs, suggesting that a substantial proportion of the breastfed newborns may be exposed to drugs in milk [4]. While the majority of drugs taken by nursing mothers did not appear to have a clear negative impact on the child, some case reports have shown severe infant poisoning examples [3,4,14]. Pharmacovigilance must keep up with the constantly changing regulatory environment, reporting, and new treatments and technologies being marketed. Persistent awareness and continued reporting are hereby necessary to meet these requirements [15]. With our study, we can conclude that there is a fluctuating but generally rising trend over the years, which means continued reporting is perceived to be important.

The patterns retrieved in the FAERS database demonstrated that 68.4% of the infants with reported ADEs were under 2 months old, 5.5% had life threatening adverse events, and 3.6% died, while 84.70% of the cases were categorized as serious in the FAERS. Our findings seem to confirm the relevance of the infant’s age when considering the safety of maternal medication use during breastfeeding. In a study conducted by Anderson et al. infants younger than two months old experienced most of the drug-related side effects during breastfeeding [16]. Similarly, in a more recent study by Anderson et al. 63% of the ADRs occurred in the first month, and 16% appeared in the second month [17]. In neonates, the most significant aspect was their quickly changing physiology, reflected in poor clearance in the first weeks, to months, of life [18]. Further, the renal clearance changed rapidly during infancy [19]. Consequently, there were notable variations in the toxicity and efficacy of treatments due to the functional maturity, development, and illness conditions of the newborn or infant [18,20].

We also observed that more than half of the reported drugs belonged to the nervous system, followed by antineoplastic and immunomodulating agents, anti-infectives for systemic use, alimentary tract and metabolism agents, and respiratory system treatments (Figure 3). In the earlier mentioned study by Anderson et al. 70% of ADRs during breastfeeding were also caused by central nervous system (CNS) active drugs, such as opioids, antidepressants, anticonvulsants, antipsychotics, lithium, or sedatives. CNS active drugs were followed by iodine (6%), antimicrobials (6%), and yellow fever vaccine (6%) [17]. In essence, our current observations using the FAERS database are in line with these results.

Another noteworthy observation is that antineoplastic and immunomodulating agents were the second most frequently reported drugs used by breastfeeding mothers. Antineoplastic and immunomodulating agents are a group of drugs that are prescribed during pregnancy and the lactation period based on the benefit/risk ratio. The reason the second most-reported drug group is antineoplastic and immunomodulating agents may be the increase in the frequency of cancer and immune system diseases in the last decades. Whether the warnings about these newer drugs are clear and comprehensible and the education about drug safety during pregnancy and breastfeeding is sufficient throughout the world should be re-evaluated in detail. The latest opinion of the FDA confirms the need for the continued education of healthcare providers about the use of prescription drugs during lactation [21].

Related to the second aim (FAERS to LactMed^®^), it was observed that the FAERS and LactMed^®^ databases have both similarities as well as differences in terms of the nature or type of the adverse drug events or reactions (Appendix A). Besides causality assessment-related differences (ADE versus ADR), this likely relates to various factors such as maternal and neonatal PKs, the time course of maternal therapy, dosing interval, duration of exposure, milk production, and daily milk volume intake. The standardization between both ‘systems’ is also different, likely because of the different settings, aims, and workflow. The FAERS database receives a significant portion of data by a regulatory type of communication, so it is likely largely standardized by medDRA approaches and word choice. In contrast, for PubMed and LactMed^®^, data are derived from the scientific literature and fully referenced. A peer review panel hereby reviews the data to assure scientific validity and currency. In our reading, the different approaches and differences in data standardization between (FAERS, ADE) and (LactMed^®^, ADR) likely explain these differences. In this way, the FAERS likely serves as a ‘sensitive’ tool, while the Lactmed^®^ rather serves as a more ‘specific’ tool.

Drug excretion in breast milk depends on different factors such as milk composition, drug properties, and transport mechanisms. The drug’s affinity to milk, pH, ionization, molecular weight, protein binding affinity, and lipid solubility affect the drug’s concentration in milk. Most of the drugs are transported into mammary blood capillaries via passive transport, but some of them are transported through different mechanisms like active transport, lipid co-transport, and transcytosis [1,5,22]. Due to the significant physiological changes linked to pregnancy such as the greater organ blood flow, higher circulating volume, and altered function of some drug-metabolizing enzymes, understanding the PKs of nursing mothers and the rate and amount of distribution of a given drug into their milk has drawn more attention in recent years [4,22]. In addition to PKs, it is known that alterations in the mother’s pharmaco-genotype can also impact metabolic or elimination pathways and potentially increase the drug exposure of their breastfed infant [5]. Measuring drug concentrations in breast milk to quantify exposure increases the understanding of the likelihood of potential side effects [22]. The FDA has issued guidelines requesting pharmaceutical companies address the potential impacts of maternal drug exposure, drug levels in breast milk, infant feeding, and drug effects on milk production [22,23]. Along the same line, we understood that the European Medicines Agency is revising their guideline on pregnancy and lactation labelling [24]. In addition to drug concentrations in breast milk, the maternal drug dose, milk-to-(maternal) plasma concentration ratio (M/P ratio), the time course of maternal therapy, dosing interval, duration of exposure, and daily milk volume intake are different factors affecting the likelihood of adverse events in nursing infants [4,5]. Only a small number of studies have evaluated the plasma levels of the newborns, and there is still a lack of information regarding the risk of drug exposure and the transfer of drugs into breastfed infants. In the literature, there are different examples of therapeutic drug monitoring as a successful method to analyze drug exposure in breastfed infants [25,26,27].

Even though the FAERS database presents valuable, real-world data for lactation-related ADEs, there is no assurance that the ADEs were caused by the actual API or substance. The cause of an event could have been another drug, an underlying illness, or just being a time-association event. Consequently, a causality evaluation has value to generate more guidance and support for sharing decision-making. Although the regulatory framework for causality evaluation and reporting in neonates is comparable to other populations, determining causation in neonates is still challenging [28]. Anderson et al. (2016) and Yalcin et al. (2024) have indicated that the causality tool provided by the WHO Uppsala Monitoring Center (WHO-UMC) and Naranjo algorithm were insufficient in their performance to document neonatal causality in a reliable and sufficient manner [17,28]. Leopoldino et al. (2023) expressed that the Du algorithm (modified Naranjo algorithm for neonates by Du et al. (2013) [29]) demonstrated good sensitivity for identifying ADRs as definite, proving to be a more appropriate tool for the neonatal clinical routine [30].

Related to the assessment of other aspects such as seriousness and severity, ADEs were categorized according to their seriousness as “serious” or “non-serious”, based on the FDA guidance. Related to severity, the Neonatal Adverse Event Severity Score (NAESS) was developed and validated for the severity assessment [28,31]. Even though it is a time-consuming procedure, a severity assessment might be useful in revealing the impact of ADRs.

### 3.2. Strengths and Limitations

Some strengths can be considered. First, we used the online, freely available FAERS database, offering a large amount of real-world data that could be used to examine the occurrence and possible relationships between drug exposure during lactation and drug-related ADEs in infants. Compared to smaller, single-center research databases or registries, the FAERS database contains much more data available from a worldwide population. Second, to compare against the FAERS ADEs data, we chose LactMed^®^, a renowned, freely available, and international reference on drug safety during lactation.

However, some limitations should also be considered. First, raw data were extracted from the FAERS database, which consists of reports of human adverse events submitted by the pharmaceutical industry, healthcare professionals, and consumers. These reports are made publicly available by the FAERS Public Dashboard, through an online platform. Even though it is easily accessible to everyone, the system only contains spontaneous, potentially incomplete, and duplicate reports. These reports include the reporter’s observations and opinions, but the content has not been externally confirmed. In addition, there is no guarantee that the adverse event reported was actually due to the suspected substance, as a formal causality assessment is lacking (ADE versus ADR). Therefore, the FAERS database should rather be perceived as an exploratory, signal detection tool to identify ADEs or potential ADRs, at the cost of specificity [10]. So, this study does not provide evidence on the causal relationships between drug exposure during lactation and adverse events in nursing infants.

### 3.3. New Methods for Future Strategies

Different in vitro and in vivo animal studies have been developed to determine the drug concentrations in breast milk [32]. While reports of a similar hormonal regulation of milk production have been made for several species, high-quality data describing species’ differences are still not sufficiently reported. Even though animal data could provide information in some manner, the Pregnancy and Lactation Labeling Rule (PLLR) established by the FDA suggests that if human data are available, animal data should not be used [5]. In recent years, in silico methods have gained importance for determining the drug concentrations in breast milk. Physiologically based pharmacokinetic (PBPK) modeling and population pharmacokinetic (popPK) modeling are useful methods for screening and determining the drug concentrations in breast milk and for estimating infant risk through breastfeeding [33,34]. Also, methods for estimating the M/P ratio that rely on the quantitative structure–activity relationship (QSAR) show promise [5,35]. However, at the very least, validation of the results is important [5,27]. In addition to these methods, machine learning models are also used for predicting xenobiotics’ transfer from maternal plasma to human milk. Even though the results are encouraging, more research is required to increase and confirm these regression models’ accuracy [36]. In the future, we are planning to use multiple reporting systems (e.g., Eudravigilance (Europe), DAEN (Australia), Vigibase (WHO), etc.) in addition to FAERS and compare the obtained data for a comparative analysis.

### 3.4. Practical Recommendations for Healthcare Professionals and Breastfeeding Mothers

Many of the publicly accessible and currently available information sources regarding drug safety during breastfeeding may be out-of-date, incorrect, or incomplete. As a result, it is crucial that trustworthy, scientifically verified, and frequently updated information sources like LactMed^®^ exist. Educating healthcare providers about recent advancements including therapeutic drug monitoring, novel treatment approaches, and emerging technologies is crucial [5,21,25]. In this manner, breastfeeding women can receive more useful information about drug safety during lactation [5]. Considering that there are not enough clinical studies in the literature on the safe use of drugs during breastfeeding, it is thought that increasing cooperation between academia, the industry, and regulatory institutions to facilitate breastfeeding studies will be beneficial in obtaining new scientific data [21]. Since we can obtain limited demographic and clinical data from the FAERS database, this makes it difficult to draw firm conclusions regarding safe drug use during lactation. By encouraging and raising awareness among healthcare professionals about adverse event reporting, and an emphasis on complete and detailed demographic and clinical data (e.g., maternal age, breastfeeding duration, infant medical history, and co-morbidities, etc.) entry into the system during reporting as important for pharmacovigilance, it is believed that more detailed and useful resources for safe drug use will be obtained.

## 4. Materials and Methods

### 4.1. Study Design

We performed an observational, cross-sectional, comparative study using the FAERS database. We hereby identified all lactation-related ADE reports entered between 1 January 2001 and 31 March 2024 (latest updated version in FAERS at the time of data extraction, 29 July 2024, and performed by the first author). The information extracted from the FAERS database was subsequently compared to information found in LactMed^®^. No ethical approval or patient consent were required.

### 4.2. Data Extraction from the FAERS Database

In the FAERS database, we searched for ADEs reported in neonates or infants associated with lactation-related drug exposure. To do so, we defined the study population, as either neonates (0–1 month), or 1 month–2 years old infants. Second, exposure routes were selected as (a) exposure via breast milk, (b) breastfeeding, (c) intoxication by breastfeeding, or (d) maternal exposure during breastfeeding. Reports on fetal exposure data during pregnancy, as well as reports on mother’s milk characteristics such as odor and discoloration were hereby excluded, and duplications were checked. The selected data were extracted from the FAERS database. As descriptive statistics, number and percentage values for categorical variables were given. All analyses were carried out in the Microsoft Excel version 16.89 software.

### 4.3. Data Handling, Analysis, and Comparison to the LactMed^®^ Database

Because of the occasionally missing data of the FAERS database, the maximum weight limit was set at 15 kg. Weight information indicated as lb was converted to kg via the “1 lb = 0.45 kg” formula [37]. Cases with both missing age and weight information had to be removed.

The reporting trends (annual number), specific infant outcomes (ADE) as mentioned in the ADE reports, and the number of ADEs according to the continent were described using absolute numbers and percentages. The ADE reports were classified using the generic name of the active pharmaceutical ingredient (API) involved and applying the first level of the Anatomical Therapeutic Chemical (ATC) classification system. Based on this list, the reported lactation-related ADEs for the top 5 drugs of the 5 most common ATC classes—25 APIs—were ranked according to the number of reports. Subsequently, these APIs were screened in the LactMed^®^ database, and findings on the type of events reported in both databases were compared in a qualitative way.

## 5. Conclusions

In total, 2628 ADEs were obtained from the FAERS database, with increased reporting over time. In the FAERS database, 68.4% of the patients were under 2 months old, 5.5% had life threatening ADEs, and 3.6% died, while 84.70% of the cases were categorized as serious. By comparing the FAERS pharmacovigilance database with the LactMed^®^ resource, we observed both differences as well as similarities in the type of events. In our reading, the different approaches and differences in data standardization between (FAERS, ADE) and (LactMed^®^, ADR) likely explain these differences. In this way, the FAERS likely serves as a ‘sensitive’ tool, while the LactMed^®^ rather serves as a more ‘specific’ tool to inform clinicians. The FAERS database is useful for obtaining exploratory information about ADEs during lactation to increase the knowledge about drug safety during breastfeeding and the awareness of possible risks in nursing infants, while LactMed^®^ translates all available information into guidance.

## Figures and Tables

**Figure 1 pharmaceuticals-17-01654-f001:**
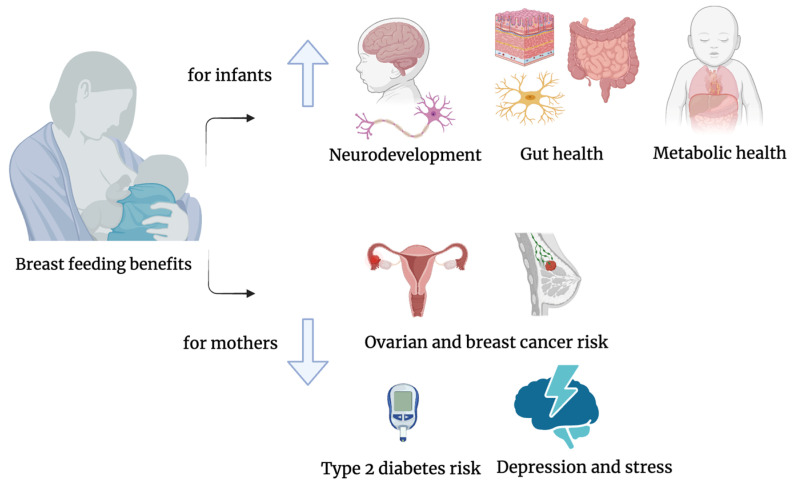
Illustrations on benefits of breastfeeding to mother and nursing infant [2,3].

**Figure 2 pharmaceuticals-17-01654-f002:**
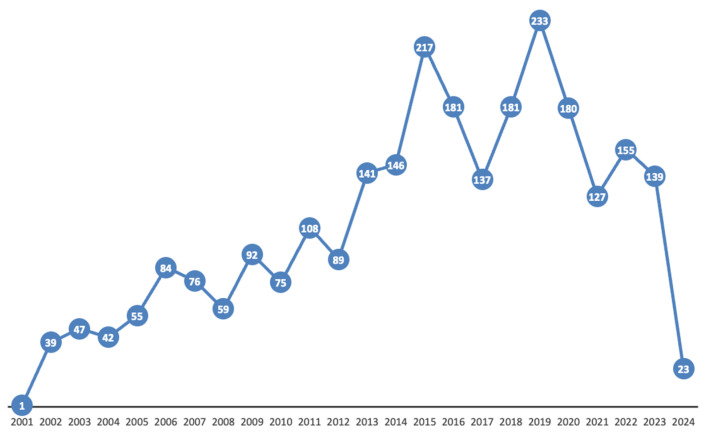
Trend in the number of lactation-related adverse events in the FAERS database (1 January 2001–31 March 2024).

**Figure 3 pharmaceuticals-17-01654-f003:**
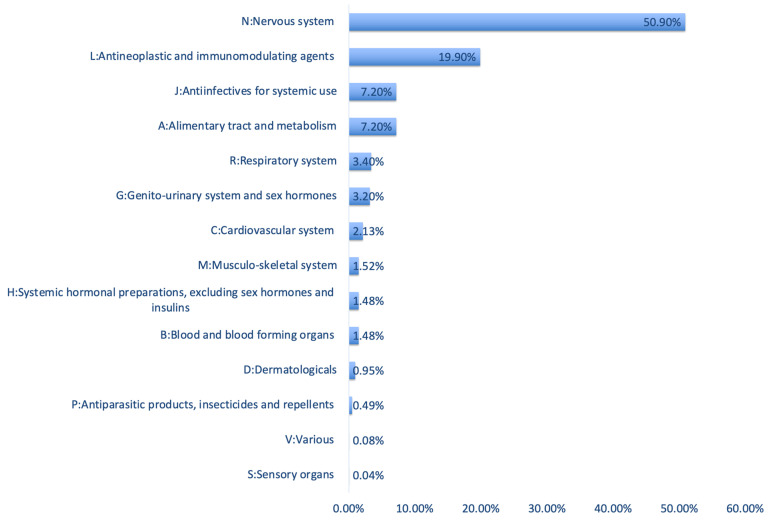
Overview of the reported drugs obtained from the FAERS database according to first level of ATC categories.

**Figure 4 pharmaceuticals-17-01654-f004:**
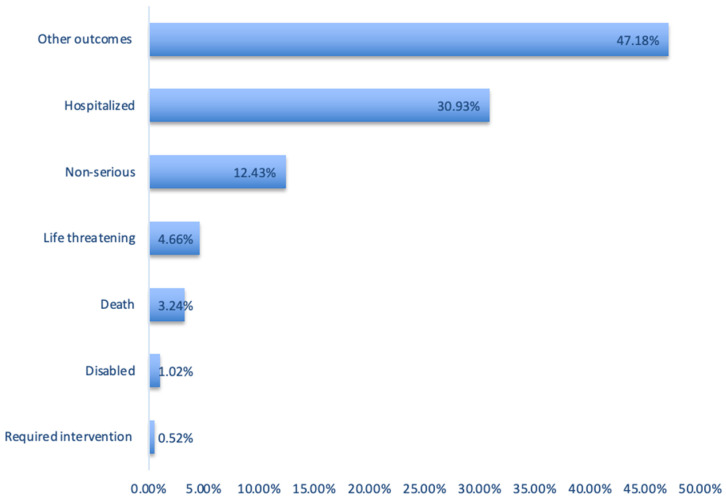
Distribution of the infant outcome categories among lactation-related adverse events in the FAERS database.

**Figure 5 pharmaceuticals-17-01654-f005:**
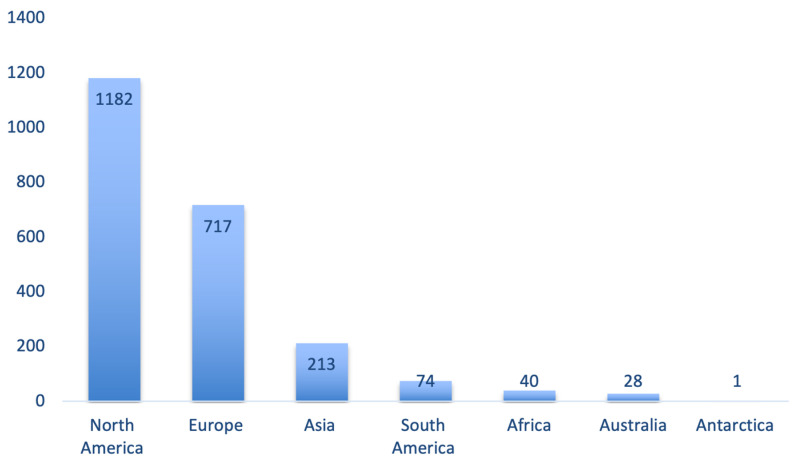
Number of lactation-related adverse events in the FAERS database according to geographical region.

**Figure 6 pharmaceuticals-17-01654-f006:**
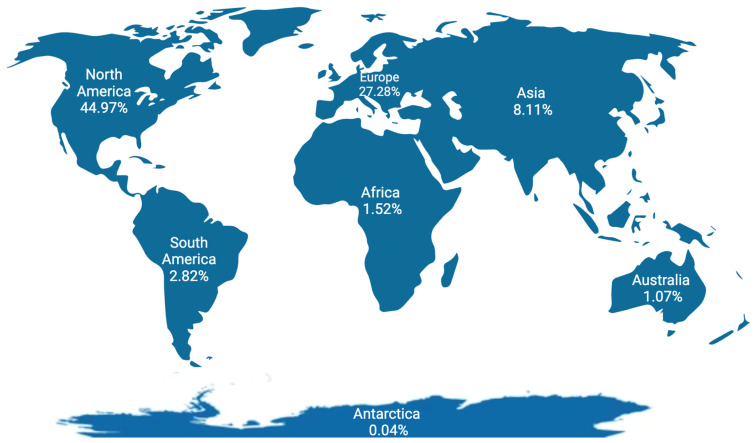
Distribution of lactation-related adverse events in the FAERS database according to continents.

**Table 1 pharmaceuticals-17-01654-t001:** Top 5 ATC classes and top 5 active pharmaceutical ingredients obtained from the FAERS database between 1 January 2001 and 31 March 2024 (the percentage refers to the percentage for a given active pharmaceutical ingredient based on the total number of the ATC class and the specific numbers of adverse events).

ATC Class	Active Pharmaceutical Ingredient	Percentage *
N: Nervous system	Buprenorphine	8.14%
	Lamotrigine	7.23%
	Levetiracetam	5.14%
	Acetaminophen	4.91%
	Nicotine	3.88%
L: Antineoplastic and immunomodulating agents	Certolizumab pegol	6.55%
	Adalimumab	3.92%
	Etanercept	2.25%
	Infliximab	2.09%
	Tacrolimus	1.07%
A: Alimentary tract and metabolism	Insulin	3.58%
	Omeprazole	0.53%
	Ondansetron hydrochloride	0.49%
	Mesalamine	0.46%
	Metformin hydrochloride	0.34%
J: Anti-infectives for systemic use	Zanamivir	1.26%
	Amoxicillin/Clavulanic acid	0.95%
	Tenofovir disoproxil fumarate	0.88%
	Lamivudine	0.69%
	Emtricitabine\Tenofovir	0.53%
R: Respiratory system	Omalizumab	0.84%
	Cetirizine hydrochloride	0.53%
	Fluticasone propionate/Salmeterol xinafoate	0.46%
	Elexacaftor\Ivacaftor\Tezacaftor	0.34%
	Budesonide	0.26%

* % of the total number of adverse events of a specific ATC class.

## Data Availability

The FAERS is an open data source, and we have provided all individual hits and results as an Excel file in the Appendix A.

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
