# Peer review of "Drug Safety During Breastfeeding: A Comparative Analysis of FDA Adverse Event Reports and LactMed®"

_pharmaceuticals, 2024, doi:10.3390/ph17121654_

Round 1
Reviewer 1 Report
Comments and Suggestions for Authors
I read with interest the paper titled "Drug Safety During Breastfeeding: A Comparative Analysis of FDA Adverse Event Reports and LactMed®"
I have a few comments and questions that could improve the manuscript.
1. Since authors used total of percentages in table 1, this is hard to interpret. I suggest to use relative frequencies n(%) of all cases, to make the comparison between groups easier to read and much more informative.
2. Why are the terms present in Table 2 very different from one database to another? It's very hard to find the same term in both databases, and somehow this is intriguing. Moreover, authors should define if the dictionary of medical terms used is the same (Meddra maybe?) or different.
3. The Table 2 is too long. I suggest to move to Appendix, since this is informative, but doesn't add much to the discussion.
4. I feel that authors failed on fulfilling the objective. When reading the conclusion, this is more a descriptive of both databases, then a comparison. Why are them so different? What makes them different? Whats the point of having two databases like those? What's the arising knowledge from integration such databases?
5. Whats the main conclusions of your study, rather that some percentages? Remebering that the objective was to "to describe the pattern of ADEs reported in the FAERS database, and to compare the number and type of ADE reports in infants related to breastfeeding to the information on side effects in the lactating infant as reported in the LactMed® database.", I feel that are much more to be stated. Authors claim both differences as well as similarities in type of events, but this should be explored (within the manuscript and of course in the conclusion afterwards).
Author Response
I read with interest the paper titled "Drug Safety During Breastfeeding: A Comparative Analysis of FDA Adverse Event Reports and LactMed®". I have a few comments and questions that could improve the manuscript.
Since authors used total of percentages in table 1, this is hard to interpret. I suggest to use relative frequencies n(%) of all cases, to make the comparison between groups easier to read and much more informative.
Answer: Thank you for your valuable comments. We changed it to relative frequencies n(%) of all cases.
Why are the terms present in Table 2 very different from one database to another? It's very hard to find the same term in both databases, and somehow this is intriguing. Moreover, authors should define if the dictionary of medical terms used is the same (Meddra maybe?) or different.
Answer: In FAERS ADEs can be reported by healthcare providers, patients, or pharmaceutical companies, while more commonly based on MedDRA classification. ADEs categorized by FAERS in accordance with the declaration of these stakeholders were included in the analysis without modification to preserve the originality of the reporting. However, as is known, ADE/ADRs are given as a narrative review in LactMed, so ADEs were extracted by preserving the originality. Dictionary of medical terms is not used to preserve the original data. In this way, we assume that the standardization between both ‘systems’ is different, likely because of the different settings, aims and workflow.
The Table 2 is too long. I suggest to move to Appendix, since this is informative, but doesn't add much to the discussion.
Answer: Thank you for your suggestion. It will be moved to Appendix as a supplementary file.
I feel that authors failed on fulfilling the objective. When reading the conclusion, this is more a descriptive of both databases, then a comparison. Why are them so different? What makes them different? Whats the point of having two databases like those? What's the arising knowledge from integration such databases?
Your reflection is well taken, and somewhat links to your question 2. The standardization between both ‘systems’ is different, likely because of the different settings, aims and workflow as e.g. the FAERS database receives a significant portion of data by regulatory type of communication, so likely largely standardized by medDRA approaches. In contrast, for PubMed and LactMed, data are derived from the scientific literature and fully referenced. A peer review panel hereby reviews the data to assure scientific validity and currency. In our reading, the different approaches p, data standardization, and on the differences between ADE (FAERS) and ADR (LactMed) likely explains the differences.
In this way, we have compared both datasets, and hereby observed indeed relevant differences, so that the FAERS likely rather serves as a ‘sensitive’ tool, while the Lactmed rather serves and as ‘specific’ tool when reading on ADEs or ADRs in both datasets.
What’s the main conclusions of your study, rather that some percentages? Remebering that the objective was to "to describe the pattern of ADEs reported in the FAERS database, and to compare the number and type of ADE reports in infants related to breastfeeding to the information on side effects in the lactating infant as reported in the LactMed® database.", I feel that are much more to be stated. Authors claim both differences as well as similarities in type of events, but this should be explored (within the manuscript and of course in the conclusion afterwards).
We understood your comment as a continuation of the previous comment, and have added reflections on this.

Reviewer 2 Report
Comments and Suggestions for Authors
The close connection between the mother and child, remains for a fairly long time after the birth, through the mother’s milk. During the early part of the child’s life nutrition is supplied by the mother and her milk gives numerous advantages to both, her child and herself. Unfortunately milk remains the medium for transfer of unwanted substances along with essential ones for the growth and well-being of the child. For long the consumption of food and medicines during lactation has been subject to intense study for the aforementioned reasons.
A large number of medicines have been restricted to lactating mothers for fear of effects on the child, hence the needs for adequate information on every drug that the mother is likely to take during lactation. The current paper has used two important databases (U.S. Food and Drug Administration (FDA) Adverse Event Reporting System (FAERS) and LactMed® a part of the TOXNET system) for analyzing the reported adverse events on the nursing child due to medication used by the mother. These two pharmacovigilance databases attempt to record each and every adverse event noted in a nursing child and try to correlate them with her medications.
The advantage of this method is that it captures every possible adverse event in the child, but the disadvantage is that it does so, without causality analysis. Thus a large number of medications risk being unfairly contraindicated in nursing women, leading to the mother being denied the required medicine. The authors have done well to identify leading medications which are known to produce adverse events in the nursing children. It may be noted that what the authors have picked up are serious adverse events and not all adverse events.
It is rather surprising that some drugs like anti-cancer agents and CMS active agents feature in this list. In most parts of the world, these compounds are ‘prescription only’ drugs. One expects that the prescribing doctor would avoid prescribing these medicines to nursing women. That these drugs were prescribed without any thought of the risk to the nursing child is to say the least very strange. This paper therefore calls for renewed education of prescribing doctors or clear labelling some of these medicines.
Nicotine has been known since long, to call harmful effects, if used during pregnancy or lactation; however, it is never used under medical advice, but the use of drugs like buprenorphine is a bit shocking. The FDA classifies all drugs on the basis of their risk during pregnancy and lactation. There appears to be a need for educating medical professionals about this classification. The alternate solution could be labelling some of these drugs more clearly so that they will not be taken by pregnant or lactating mothers.
Comments on the Quality of English Language
In some places there is a lack of clarity of what the authors want to say, a thorough review of the grammar may be beneficial.
Author Response
The close connection between the mother and child, remains for a fairly long time after the birth, through the mother’s milk. During the early part of the child’s life nutrition is supplied by the mother and her milk gives numerous advantages to both, her child and herself. Unfortunately milk remains the medium for transfer of unwanted substances along with essential ones for the growth and well-being of the child. For long the consumption of food and medicines during lactation has been subject to intense study for the aforementioned reasons.
A large number of medicines have been restricted to lactating mothers for fear of effects on the child, hence the needs for adequate information on every drug that the mother is likely to take during lactation. The current paper has used two important databases (U.S. Food and Drug Administration (FDA) Adverse Event Reporting System (FAERS) and LactMed® a part of the TOXNET system) for analyzing the reported adverse events on the nursing child due to medication used by the mother. These two pharmacovigilance databases attempt to record each and every adverse event noted in a nursing child and try to correlate them with her medications.
The advantage of this method is that it captures every possible adverse event in the child, but the disadvantage is that it does so, without causality analysis. Thus a large number of medications risk being unfairly contraindicated in nursing women, leading to the mother being denied the required medicine. The authors have done well to identify leading medications which are known to produce adverse events in the nursing children. It may be noted that what the authors have picked up are serious adverse events and not all adverse events.
It is rather surprising that some drugs like anti-cancer agents and CMS active agents feature in this list. In most parts of the world, these compounds are ‘prescription only’ drugs. One expects that the prescribing doctor would avoid prescribing these medicines to nursing women. That these drugs were prescribed without any thought of the risk to the nursing child is to say the least very strange. This paper therefore calls for renewed education of prescribing doctors or clear labelling some of these medicines.
Nicotine has been known since long, to call harmful effects, if used during pregnancy or lactation; however, it is never used under medical advice, but the use of drugs like buprenorphine is a bit shocking. The FDA classifies all drugs on the basis of their risk during pregnancy and lactation. There appears to be a need for educating medical professionals about this classification. The alternate solution could be labelling some of these drugs more clearly so that they will not be taken by pregnant or lactating mothers.
Answer: Thank you for your positive feedback and pointing this out. We added a paragraph on this aspect “Another noteworthy point among our results is that we found antineoplastic and immunomodulating agents are the second most frequently reported drugs which is used by breastfeeding mothers. Antineoplastic and immunomodulating agents are a group of drugs that are prescribed during pregnancy and lactation period based on the benefit/risk ratio. The reason for second most reported drug group is antineoplastic and immunomodulating agents, may be the increase in the frequency of cancer and immune system diseases in the last decades. However, this situation may also be due to medical professionals thinking that these drugs will not harm the infants because the pregnancy period is over. Whether the warnings about drugs are clear and comprehensible enough and the education about drug safety during pregnancy and breastfeeding is sufficient throughout the world should be re-evaluated in detail. The last opinion of FDA also expresses the need for education of healthcare providers about use of prescription drugs during lactation in the absence of adequate clinical studies (Koenig et al. Healthcare providers' use of a concise summary to prescribe for lactating patients. Res Social Adm Pharm. 2024 May;20(5):531-538.)” to the discussion section.

Reviewer 3 Report
Comments and Suggestions for Authors
The work consists of an analysis of the FDA adverse reaction surveillance database (FAERS) and a specific system linked to data on drug passage through breast milk.
The title is a bit long, but clear, descriptive and acceptable.
The abstract is of adequate length, divided into sections, each of which presents the main characteristics of the manuscript, showing specific numerical results of interest.
The introduction makes an adequate review of the contexts linked to breastfeeding, its benefits for the health of the child and the mother, and the possibilities of drug passage and its effects.
It is short, but sufficient to justify the work and express its objectives.
The materials and methods are described superficially and require further description. The software used to perform the processing is not mentioned. The calculated parameters and formulas used or specific tests for comparison are not explained.
The computer equipment used is not mentioned.
The results are expressed clearly, succinctly, with adequate parameters, using good quality tables and graphs, and adequate dispersion measures when appropriate. No significant biases are identified.
The discussion is somewhat extensive, and addresses the main results, comparing them with the bibliography and analyzing their possible implications and foundations. The limitations of the work are adequately analyzed.
The conclusion is somewhat extensive, and repeats data from results, when in truth it should be reduced to concrete statements that are a consequence of the results already expressed.
Author Response
The work consists of an analysis of the FDA adverse reaction surveillance database (FAERS) and a specific system linked to data on drug passage through breast milk.
The title is a bit long, but clear, descriptive and acceptable.
The abstract is of adequate length, divided into sections, each of which presents the main characteristics of the manuscript, showing specific numerical results of interest.
The introduction makes an adequate review of the contexts linked to breastfeeding, its benefits for the health of the child and the mother, and the possibilities of drug passage and its effects.
It is short, but sufficient to justify the work and express its objectives.
The materials and methods are described superficially and require further description. The software used to perform the processing is not mentioned. The calculated parameters and formulas used or specific tests for comparison are not explained.
The computer equipment used is not mentioned.
Answer: Thank you for your valuable comments. These sentences have been added in line with your suggestion: “As descriptive statistics, number and percentage values for categorical variables were given. All analysis were carried out in the Microsoft Excel version 16.89 software.”
The results are expressed clearly, succinctly, with adequate parameters, using good quality tables and graphs, and adequate dispersion measures when appropriate. No significant biases are identified. The discussion is somewhat extensive, and addresses the main results, comparing them with the bibliography and analyzing their possible implications and foundations. The limitations of the work are adequately analyzed.
we have adapted the discussion, along the suggestions provided by the reviewers, but value your positive assessment
The conclusion is somewhat extensive, and repeats data from results, when in truth it should be reduced to concrete statements that are a consequence of the results already expressed.
We have adapted (shortened) the conclusions.

Reviewer 4 Report
Comments and Suggestions for Authors
The study commendably employs the comprehensive FAERS database to explore a crucial yet often overlooked area: the safety of medications during breastfeeding. The comparison with LactMed® reveals insightful differences between spontaneous reporting systems and evidence-based resources, emphasizing the importance of ongoing reporting and raising awareness of risks to breastfed infants.
However, the study could be enhanced by addressing several limitations:
Data and Analysis:
Limited Demographics and Clinical Data: Without information on maternal age, breastfeeding duration, infant medical history, and co-morbidities, it’s challenging to identify specific risk factors and draw robust conclusions.
Lack of Statistical Analysis: The absence of statistical tests limits the assessment of findings’ significance and meaningful comparison between FAERS and LactMed® data.
Figure and Presentation:
Figure 4: Breaking down the “other outcomes” category into subcategories would offer a clearer understanding of the reported outcomes’ diversity and frequency.
Figure 5: A map or further explanation of the geographical distribution of reports would enhance comprehension and interpretation.
Discussion and Future Perspectives:
Confounding Factors: The study could explore potential confounders like maternal conditions or other medications that might contribute to ADEs. Analyzing data while controlling for these factors would strengthen findings.
Improving Reporting Systems: Strategies to enhance the accuracy and completeness of reporting systems like FAERS, such as increasing healthcare professional involvement or implementing incentives, should be discussed.
Actionable Recommendations: Practical recommendations for healthcare professionals and breastfeeding mothers based on the findings, including guidance on using resources like LactMed® and considering therapeutic drug monitoring, should be provided.
Additional Considerations:
Causality Assessment: While acknowledging FAERS’ limitations in causality assessment, discussing the implications and potential methods to address this limitation would be valuable.
Comparison with Other Databases: Comparing findings with other reporting systems could provide a broader perspective on ADEs during breastfeeding.
In summary, the study significantly contributes to understanding drug safety during breastfeeding. Addressing the mentioned limitations and incorporating suggested enhancements would strengthen the study and its impact on clinical practice and policy development.
Author Response
The study commendably employs the comprehensive FAERS database to explore a crucial yet often overlooked area: the safety of medications during breastfeeding. The comparison with LactMed® reveals insightful differences between spontaneous reporting systems and evidence-based resources, emphasizing the importance of ongoing reporting and raising awareness of risks to breastfed infants.
However, the study could be enhanced by addressing several limitations:
Data and Analysis:
Limited Demographics and Clinical Data: Without information on maternal age, breastfeeding duration, infant medical history, and co-morbidities, it’s challenging to identify specific risk factors and draw robust conclusions.
Answer: To emphasize the importance of limited demographics and clinical data, we added a paragraph “Since we can obtain limited demographic and clinical data from the FAERS database, this makes it difficult to draw firm conclusions regarding safe drug use during lactation, even more because events are reported, not neccesary causal. By encouraging and raising awareness among healthcare professionals about adverse event reporting and emphasis on complete and detailed demographic and clinical data (e.g. maternal age, breastfeeding duration, infant medical history, and co-morbidities etc.) entry into the system during reporting is important for pharmacovigilance, it is believed that more detailed, useful and robust resources for safe drug use will be obtained.” to the discussion section according to your valuable suggestions.
Lack of Statistical Analysis: The absence of statistical tests limits the assessment of findings’ significance and meaningful comparison between FAERS and LactMed® data.
Answer: Thank you for your valuable comments. Due to the retrospective and descriptive study design we have limited data and variables for extentive analysis. Eventhough, to describe the analysis we made, these sentences have been added: “As descriptive statistics, number and percentage values for categorical variables were given. All analysis were carried out in the Microsoft Excel version 16.89 software.”
Figure and Presentation:
Figure 4: Breaking down the “other outcomes” category into subcategories would offer a clearer understanding of the reported outcomes’ diversity and frequency.
Answer: As you can see from the raw data which can be found in the supplementary excel file, in the FAERS database, outcomes are classified by the FDA as: required intervention, disabled, death, life threatening, non-serious, hospitalized and other outcomes. Although we wanted to make a detailed discussion by dividing “other outcomes” into subcategories, the subcategorization process could not be carried out because there was no additional, detailed information in the database.
However, we will discuss this lack of information is important in the advantage and disadvantage section of our study and that if detailed information was included, it would contribute more to us in terms of interpreting the results more strikingly.
Figure 5: A map or further explanation of the geographical distribution of reports would enhance comprehension and interpretation.
Answer: In order to make it easier to understand and more eye-catching, geographical data has been summarized on a world map and added as a new figure (Figure 6).
Discussion and Future Perspectives:
Confounding Factors: The study could explore potential confounders like maternal conditions or other medications that might contribute to ADEs. Analyzing data while controlling for these factors would strengthen findings.
Answer: As you mentioned in your comment, potential confounders like maternal conditions or other medications that might contribute to ADEs are quite important. The FAERS database does not contain sufficient demographic and clinical data about the mother and the neonate.
Improving Reporting Systems: Strategies to enhance the accuracy and completeness of reporting systems like FAERS, such as increasing healthcare professional involvement or implementing incentives, should be discussed.
Answer: We added a paragraph “Since we can obtain limited demographic and clinical data from the FAERS database, this makes it difficult to draw firm conclusions regarding safe drug use during lactation. By encouraging and raising awareness among healthcare professionals about adverse event reporting and emphasis on complete and detailed demographic and clinical data (e.g. maternal age, breastfeeding duration, infant medical history, and co-morbidities etc.) entry into the system during reporting is important for pharmacovigilance, it is believed that more detailed, useful and robust resources for safe drug use will be obtained.” to the discussion section according to your valuable suggestions.
Actionable Recommendations: Practical recommendations for healthcare professionals and breastfeeding mothers based on the findings, including guidance on using resources like LactMed® and considering therapeutic drug monitoring, should be provided.
Answer: Thank you for your suggestions. We added a paragraph “Many of the publicly accessible and currently available information sources regarding the safe use of drugs while breastfeeding may be out-of-date, incorrect or incomplete. As a result, it is crucial that trustworthy, scientifically verified information sources like LactMed exist and are updated frequently. Educating healthcare providers about recent advancements including therapeutic drug monitoring, novel treatment approaches, and emerging technologies is crucial. In this manner, nursing women can receive more helpful information on the safe use of drugs. Considering that there are not enough clinical studies in the literature on the safe use of drugs during breastfeeding, it is thought that increasing cooperation between academia, industry and regulatory institutions to facilitate breastfeeding studies will be beneficial in obtaining new scientific data. By encouraging and raising awareness among healthcare professionals about adverse event reporting and emphasis on complete data entry into the system during reporting is important for pharmacovigilance, it is believed that more detailed and useful resources for safe drug use will be obtained.” to the discussion section according to your valuable suggestions.
Additional Considerations:
Causality Assessment: While acknowledging FAERS’ limitations in causality assessment, discussing the implications and potential methods to address this limitation would be valuable.
Answer: Potential methods (Du algorithm) to address the lack of causality assessments were discussed briefly in the discussion section, but we also added this information in the conclusion section to make the subject more clear.
Comparison with Other Databases: Comparing findings with other reporting systems could provide a broader perspective on ADEs during breastfeeding.
Answer: As you mentioned in your valuable comment to strengthen our work, it would be very useful to use data from more than one pharmacovigilance database.These sentences are added to the manuscript: “In the future, we are planning to use multiple reporting systems (e.g. Eudravigilance (Europe), DAEN (Australia), Vigibase (WHO) etc.) in addition to FAERS and compare the obtained data for a comparative analysis.”
In summary, the study significantly contributes to understanding drug safety during breastfeeding. Addressing the mentioned limitations and incorporating suggested enhancements would strengthen the study and its impact on clinical practice and policy development.
Thank you for your positive feedback and fruitful suggestions.

Round 2
Reviewer 1 Report
Comments and Suggestions for Authors
The authors addressed all my comments. Accept in the current form.
Reviewer 4 Report
Comments and Suggestions for Authors
Thank you for your revision and I am happy to recommend the manuscript to be considered for publication in its current version.